# Induction of Cell Death in Human A549 Cells Using 3-(Quinoxaline-3-yl) Prop-2-ynyl Methanosulphonate and 3-(Quinoxaline-3-yl) Prop-2-yn-1-ol

**DOI:** 10.3390/molecules24030407

**Published:** 2019-01-23

**Authors:** Mixo Aunny Sibiya, Lerato Raphoko, Dikgale Mangokoana, Raymond Makola, Winston Nxumalo, Thabe Moses Matsebatlela

**Affiliations:** 1Department of Biochemistry, Microbiology and Biotechnology, School of Molecular and Life Sciences, University of Limpopo, Sovenga 0727, South Africa; aunnymixo@gmail.com (M.A.S.); dikgalemangokoana@gmail.com (D.M.); makolaraymond4@gmail.com (R.M.); 2Chemistry Department, School of Physical and Mineral Sciences, University of Limpopo, Sovenga 0727, South Africa; leratomaraphoko@gmail.com (L.R.); winston.nxumalo@ul.ac.za (W.N.)

**Keywords:** quinoxalines, lung cancer, apoptosis, free radicals, anticancer, antioxidant

## Abstract

Despite major advancements in the development of various chemotherapeutic agents, treatment for lung cancer remains costly, ineffective, toxic to normal non-cancerous cells, and still hampered by a high level of remissions. A novel cohort of quinoxaline derivatives designed to possess a wide spectrum of biological activities was synthesized with promising targeted and selective anticancer drug activity. Hence, this study was aimed at determining in vitro anticancer activity effects of a newly synthesized class of 3-(quinoxaline-3-yl) prop-2-ynyl quinoxaline derivatives on A549 lung cancer cells. An assessment of the quinoxaline derivatives ferric reducing power, free radical scavenging activity, cytotoxic activity, and ability to induce reactive oxygen species (ROS) production was performed using the Ferric Reducing Antioxidant Power (FRAP), 2,2-diphenyl-1-picryl-hydrazyl (DPPH), 3-[4,5-dimethylthiazole-2-yl]-2,5-diphenyltetrazolium bromide (MTT) and 2’,7’-dichlorodihydrofluorescein diacetate (H_2_DCFDA) assays, respectively. The ability of the quinoxaline derivatives to induce apoptosis in A549 cells was assessed using the Acridine Orange/Ethidium Bromide (AO/EB) and Annexin V-FITC/Dead Cell Assay. Of the four quinoxaline derivatives tested, 3-(quinoxaline-3-yl) prop-2-ynyl methanosulphate (LA-39B) and 3-(quinoxaline-3-yl) prop-2-yn-1-ol (LA-55) displayed a dose-dependent reducing power, free-radical scavenging activity, inhibition of cell viability, and stimulation of ROS production which was accompanied by induction of apoptosis in A549 lung cancer cells. None of the quinoxaline derivatives induced cell death or ROS production in non-cancerous Raw 267.4 macrophage cells. Cytotoxicity was observed in A549 lung cancer, HeLa cervical cancer, and MCF-7 breast cancer cells albeit inhibition was more pronounced in A549 cells. The results of the study suggest that 3-(quinoxaline-3-yl) prop-2-ynyl methanosulphate and 3-(quinoxaline-3-yl) prop-2-yn-1-ol induce apoptotic cell death in A549 lung cancer cells.

## 1. Introduction

Lung cancer remains among the leading causes of cancer-related death in the world affecting both males and females [1]. Non-small cell lung cancer is the foremost histological form of lung cancer and stays a principal cause of cancer-related deaths, accounting for more than one million deaths per year [2,3,4]. In patients with advanced forms of lung cancer, prognosis is very poor with a predicted survival of 8 months, even when treated with some of the best chemotherapeutic agents [5]. Despite the advancements in the discovery of chemotherapeutic agents, persistent commitment to the discovery of new anticancer agents is of significant importance. Current treatments for lung cancer remain costly, ineffective, non-specific to cancerous cells, and present with deleterious side effects [1,2,3,4]. Quinoxaline derivatives are among the most recent agents that present a new series of highly effective cancer-specific derivatives synthesized from readily available raw materials [6]. The search for more highly selective anticancer drugs with a quinoxaline core is not unique to this study, since small quinoxaline-based molecules that inhibit homologous recombination DNA repair in human cells and represent a therapeutic target in oncology were reportedly developed [7]. In addition, a novel 8-chloro-4-(4-chlorophenyl)pyrrolo[1,2-a]quinoxaline has been reported to induce selective anti-proliferative activity in breast cancer cell lines and inhibit cyclin D1 expression with a sustained induction of cell-cycle negative regulators such as p53 and p21 [8].

Quinoxalines are referred to as a benzopyrazines because their structure consists of a benzene ring and a pyrazine ring [6]. They can also be defined by their nitrogen containing heterocyclic compounds, which are capable of triggering antimicrobial activities [9]. Quinoxaline derivatives have the ability to bind DNA and this positions them among potential drugs to explore when it comes to the development of anticancer chemotherapeutic agents [6]. The pyrazine ring grants quinoxalines the ability to bind DNA and interfere with replication. Their aromatic group possesses free radical scavenging activity and can also enable quinoxaline derivatives to chemically interact with many other substances. These aromatic groups easily form adduct ions with halogens, enabling them to interact with several biological pathway intermediates including those that modulate the immune response elements mediated by macrophages [6,9]. In addition to their immune defense roles in normal physiological conditions, macrophages also play crucial roles in disease conditions such as lung cancer by regulating different steps of tumor progression and metastasis. This provides opportunities to study them in pursuit of effective cancer prevention and treatment [10]. Since classically activated macrophages produce pro-inflammatory cytokines and reactive oxygen/nitrogen species, which are crucial for host defense and the removal of lung cancer cells, it is appropriate to determine the effect of anticancer candidate drugs on the tumor microenvironment and the macrophages surrounding them [11]. Hence, in this study, Raw 264.7 macrophages are used as control cells against quinoxaline derivatives.

In this study, we document the antioxidant properties of 3-(quinoxaline-3-yl) prop-2-ynyl methanosulphate (LA-39B), and 3-(quinoxaline-3-yl) prop-2-yn-1-ol (LA-55) quinoxaline derivatives and their ability to induce cell death in A549 lung cancer cells. 

## 2. Results 

### 2.1. Chemistry

#### 2.1.1. Procedure for Synthesis of 3-(quinoxalin-3-yl)prop-2-yn-1-ol (LA-55)

Quinoxaline derivative LA-55 (Figure 1) was synthesized according to a method previously described [12,13]. We oven-dried two neck flasks equipped with a stirrer bar, quinoxalin-3-yl benzenesulfonate^1^ (1 g, 4.09 mmol), PdCl_2_(PPh_3_)_2_ (5 mol %, 0.21 mmol, 150 mg), and CuI (10 mol %, 0.41 mmol, 83 mg) were dissolved in 15 mL dry THF followed by the addition of Et_3_N (2 eq, 8.18 mmol, 1.14 mL) and propargyl alcohol (1.2 eq, 4.98 mmol, 0.29 mL). The reaction mixture was stirred at 50 °C for 18 h under nitrogen atmosphere, thereafter partitioned between EtOAc/water in (20 mL 3:1). The layers were separated and the aqueous layer was washed with EtOAc (3 × 20 mL). The combined organic layers were dried over anhydrous MgSO_4_, filtered, and concentrated. The crude product was purified by recrystallization from acetone and gave 3-(quinoxalin-3-yl)prop-2-yn-1-ol as a brown powder (451 mg, 60%), mp = 139.2 to 140.8 °C (Lit 140 to 141 °C) ^2^; δ_H_ (400 MHz, CDCl_3_, ppm) 4.610 (2H, s), 7.789 (2H, m), 8.077 (2H, m) and 8.887 (1H, s); δ_C_ (100 MHz, CDCl_3_, ppm) 51.39, 83.03, 91.90, 128.44, 129.16, 130.72, 132.01, 138.72, 141.06, 141.96, and 146.90; V_max_ (FTIR) 758, 946, 1014, 1127, 1229, 1429, 1694, 2228, 2922, 3275 cm^−1^_;_ m/z MS (ESI): MH^+^ 185.1.

#### 2.1.2. Procedure for Synthesis of 2-Methyl-4-(Quinoxalin-3-yl)but-3-yn-2-ol (LA-65C3)

Quinoxaline derivative LA-65C3 (Figure 2) was synthesized according to a method previously described [12,13]. We oven-dried two neck flasks equipped with a stirrer bar, quinoxalin-3-yl benzenesulfonate^1^ (2.04 g, 6.99 mmol), PdCl_2_(PPh_3_)_2_ (5 mol %, 0.35 mmol, 1g), and CuI (10 mol %, 0.699 mmol, 133 mg) were dissolved in 25 mL dry THF followed by the addition of Et_3_N (2 eq, 0.014 mol, 1.95 mL) and 2-methylbut-3-yn-2-ol (1.2 eq, 8.39 mmol, 0.81 mL). The reaction mixture was stirred at 50 °C for 18 h under nitrogen atmosphere, thereafter partitioned between EtOAc/water in (20 mL 3:1). The layers were separated and the aqueous layer was washed with EtOAc (3 × 20 mL). The combined organic layers were dried over anhydrous MgSO_4_, filtered, and concentrated. The crude product was purified by recrystallization from acetone and gave 2-methyl-4-(quinoxalin-3-yl)but-3-yn-2-ol as a brown powder (1.165 g, 78%), mp = 155.3 to 158.4 °C; δ_H_ (400 MHz, CDCl_3_, ppm) 1.695 (1H, s), 7.778 (2H, m), 8.076 (2H, m) and 8.871 (1H, s); δ_C_ (100 MHz, CDCl_3_, ppm) 31.07, 65.51, 79.84, 98.21, 129.14, 129.18, 130.56, 130.75, 139.01, 140.93, 141.96, and 147.09; V_max_ (FTIR) 762, 960, 1050, 1229, 1301, 1488, 1538, 2230, 2982, 3290 cm^−1^; m/z HRMS (ESI): MH^+^ 213.1020.

#### 2.1.3. Procedure for Synthesis of 3-(Quinoxalin-3-yl)propionaldehyde (LA-16A)

Quinoxaline derivative LA-16A (Figure 3) was synthesized according to a method previously described [12,13]. A mixture of Dess–Martin reagent (1.5 eq, 1.21 mmol, 515 mg) and 3-(quinoxalin-3-yl)prop-2-yn-1-ol (0.804 mmol, 148 mg)**,** were mixed in 50 mL round bottom flask containing 10 mL dichloromethane and stirred for 30 min at room temperature. Following this, aqueous 10% Na_2_S_2_O_3_.5H_2_O (10 mL) and aqueous saturated solution NaHCO_3_ (6 mL) were added into the reaction mixture and stirred for further 5 min. The layers were separated and the aqueous layer was extracted with dichloromethane (10 mL × 3). The combined organic layers were dried over anhydrous MgSO_4_, filtered, and concentrated. The crude product was purified on prep TLC 3:7 ethyl acetate/n-hexane and gave 3-(quinoxalin-3-yl)propionaldehyde as brown powder (40 mg, 28%), mp = 122.2 to 124.8 °C; δ_H_ (400 MHz, CDCl_3_, ppm) 7.873 (2H, m), 8.139 (2H, m), 9.021 (1H, s) and 9.532 (1H, s); δ_C_ (100 MHz, CDCl_3_, ppm) 88.18, 89.25, 129.46, 129.68, 131.37, 132.16, 136.41, 141.85, 124.31, 147.09, and 175.89; V_max_ (FTIR) 752, 955, 1024, 1090, 1122, 1295, 1367, 1485, 1662, 2200 cm^−1^; m/z HRMS (ES): MH^+^ 183.0552.

#### 2.1.4. Procedure for Synthesis of 3-(Quinoxalin-3-yl)Prop-2-Ynyl Methanesulfonate (LA-39B)

Quinoxaline derivative LA-39B (Figure 4) was synthesized according to a method previously described [12,13]. An oven dried two neck flask was charged with 3-(quinoxalin-3-yl)prop-2-yn-1-ol (2.71 mmol, 500 mg) and Et_3_N (3.2 eq, 8.67 mmol, 1.21 mL) in 10 mL dry THF, cooled to 0 °C, followed by drop-wise addition of MeSO_2_Cl (1.2 eq, 3.26 mmol, 0.25 mL). The reaction was maintained at 0 °C for 2.5 h under nitrogen atmosphere. The reaction was quenched by the addition of aqueous saturated solution of NaHCO_3_ (10 mL), the layers were separated and the aqueous layer was extracted with Et_2_O (10 mL × 3). The combined organic layers were dried over anhydrous MgSO_4_, filtered, and concentrated. The crude product was purified on prep TLC using 3:7 ethyl acetate/n-hexane and gave 3-(quinoxalin-3-yl) prop-2-ynyl methanesulfonate as a brown solid (423 mg, 59%), mp = 93.8 to 96.7 °C; δ_H_ (400 MHz, CDCl_3_, ppm) 3.193 (3H, s), 5.159 (2H, s), 7.795 (2H, m), 8.067 (2H, m) and 8.885 (1H, s); δ_C_ (100 MHz, CDCl_3_, ppm) 38.95, 57.12, 84.54, 86.12, 129.25, 131.00, 131.20, 137.56, 141.33, 141,97, and 146.60; V_max_ (FTIR) 523, 649, 765, 801, 938, 1008, 1172, 1355, 1490, 2953 cm^−1^; m/z HRMS (ES): MH^+^ 263.0481.

### 2.2. Determination of Reducing Power of Quinoxaline Derivatives

The FRAP assay was carried out on the four quinoxaline derivatives to evaluate their reducing power potential. Figure 5 shows the results of reducing power of quinoxaline derivatives as percentages against water as a blank. Results in Figure 5 show that these quinoxaline derivatives displayed varying degrees of reducing power in a dose-depended fashion. This trend was also observed with ascorbic acid which was used as a standard. In summary, of the four quinoxaline derivatives, LA-39B displayed the highest reducing power followed by LA-55, LA-65C3, and LA-16A.

### 2.3. Determination of Free Radical Scavenging Ability of Quinoxaline Derivatives

The DPPH assay was carried out to evaluate the free-radical scavenging abilities of the quinoxaline derivatives. Figure 6 shows the results of free radical scavenging ability of quinoxaline derivatives as percentages depicting their antioxidant properties. As determined using the DPPH assay, the quinoxaline derivatives displayed free-radical scavenging properties wherein, as the concentration increased, the free-radical scavenging abilities also increased accordingly. This trend was also observed with ascorbic acid which was used as a standard. Comparing the free-radical strengths among the four quinoxaline derivatives, LA-39B displayed the highest DPPH scavenging abilities. LA-55 was second, followed by LA-65C3, while LA-16A displayed the least DPPH-scavenging activity.

### 2.4. The Effect of Quinoxaline Derivatives LA-39B, LA-55, LA-65C3, and L-16A on Cell Proliferation on HeLa, MCF-7, A549, and Raw 264.7 Cell Lines

The ability of quinoxaline derivatives to induce cancer cell death was assessed using the MTT assay after challenging various cancer cell types with the four selected quinoxaline derivatives. Figure 7, Figure 8, Figure 9 and Figure 10, show the percentage viability of quinoxaline derivatives at different concentrations (25 µM–100 µM) in HeLa, MCF-7, A549, and Raw 264.7 cells. The results show a dose-dependent inhibition of cell viability in these cancer cell lines. LA-39B and LA-55 displayed the highest viability-inhibition abilities in all cancer cell lines with more distinctive significance in A549 lung cancer cells when compared to LA-65C3 and LA-16A which were not as effective. Figure 11 shows a comparison of cell proliferation profiles in different cell lines when treated with 25µM of quinoxaline derivatives.

### 2.5. Reduction in ROS Levels in LPS-Stimulated Raw 264.7 Cells Using LA-39B and LA-55

The two quinoxaline derivatives, LA-39B and LA-55, with high free-radical scavenging activities were tested on Lipopolysaccharide (LPS)-activated Raw 264.7 mammalian macrophages. Reactive oxygen species (ROS) profiles are used as an indicative measure of oxidative stress levels in mammalian cells [14,15,16,17]. Figure 12 shows a decrease in the amount of ROS when treated with LA-39B with an increase in the number of cells without ROS and live cell percentage of 66.13%. LA-55 also reduced ROS levels in Raw 264.7 cells, accompanied by an increase in live cell population to 62.76%. As expected, Actinomycin D was able to minimally reduce ROS and restricted live cell percentage to 41.05%.

### 2.6. Reduction in Oxidative Stress on A549 When Treated with LA-39B and LA-55

Free-radical levels in cancer cells play a crucial role in DNA stability and the ability to proliferate [14,15,16,17]; hence, a measure of ROS levels was carried out in A549 lung cancer cells after treatment with LA-39B and LA-55. Figure 13 shows an increase in the amount of ROS levels when A549 cells were treated with LA-39B and corresponding live cell viability percentage of 36.66%. LA-55 was also able to increase ROS levels in A549 cells albeit more live cells (42.66%) were recorded compared to those treated using LA-39B. As expected, Actinomycin D was able to minimally increase ROS levels in A549 with a resulting live cell percentage of 27.69%.

### 2.7. Apoptosis Analysis through Nuclear Staining on A549 Cells after Treatment with LA-39B and LA-55

Induction of apoptosis was analyzed through staining of A549 cells with Acridine Orange and Ethidium Bromide wherein double-staining with the two dyes display an orange fluorescence indicative of apoptotic cell death whereas green nuclear fluorescence designates viable cells. Figure 14 shows nuclear morphological analysis of A549 cells with corresponding resulting fluorescence after treatment with LA-39B and LA-55. Both LA-39B and LA-55 displayed characteristics of apoptosis wherein an orange fluorescence was observed in A549 lung cancer cells treated with the two quinoxaline derivatives. As expected, Actinomycin D showed orange fluorescence, resembling signs of apoptosis. 

### 2.8. Apoptosis Analysis through Nuclear Staining on A549 Cells after Treatment with LA-39B and LA-55

Apoptosis profiles are displayed in a quadratic format showing various stages of apoptosis after treatment of A549 lung cancer cells with LA-39B and LA-55. Figure 15 shows that LA-39B induced early apoptosis in A549 cells with 54.41% of cells staining positive for Annexin-V. LA-55 also induced early apoptosis in A549 cells with 30.43% of cells staining positive for Annexin-V and negative for Propidium Iodide (PI). As expected, Actinomycin D induced early apoptosis with an apoptotic profile of 39.75%.

## 3. Discussion 

Despite major advancements in the management of cancer through chemotherapy, the progressive pledge in research of discovering new treatment strategies with improved efficacy is of crucial importance [14,15]. The use of quinoxaline hybrids has been employed as a mechanism of attempting to discover treatment that will have selectivity [14]. The use of radiation along with chemotherapy are some of the effective treatment plans implemented in the control of cancer progression. These types of treatment options lead to an increase of intracellular ROS, especially in non-cancerous cells. These high levels of ROS contribute to the killing of cancer cells [16,17,18,19,20]. The imbalance in ROS levels due to certain treatments has been associated with side effects such as heart failure [21]. Therefore, the present study aimed at exploring a treatment strategy that will not only promote killing and inhibition of cancer cell proliferation, but will also be able to offer a protective effect on normal cells.

Antioxidant properties have been identified for their preventive action against cancer through a number of cellular processes which include apoptosis [22]. In this study, quinoxaline derivatives, LA-39B, and LA-55, were evaluated for antioxidant properties and ability to induce cell death in A549 lung cancer cells and non-cancerous Raw 264.7 mammalian cells. The decrease in oxidative stress observed in LPS-stimulated Raw 264.7 cells when treated with these quinoxaline derivatives may aid in the protection and enhancement of healthy immune cells by reducing the levels of ROS responsible for onset of tumorigenesis [23]. In contrast to this effect, in cancer cells oxidative stress is associated with the regulation of apoptosis. Oxidative stress is one of the pivotal events involved in the regulation of apoptosis [24]. In this study, an increase in oxidative stress in A549 cells was observed upon treatment with LA-39B and LA-55. An increase in oxidative stress is one of the therapeutic methods employed in radiation and chemotherapy. Oxidative stress forms part of optimum processes targeted in the inhibition of cancer cell proliferation [18]. Drugs that have the ability to control ROS may provide an extensive advantage in induction of cancer cell death [25,26,27]. 

Apoptosis is a programmed and regulated process in induction of cell death which is characterized by numerous cellular and molecular changes [28]. An apoptotic cell is characterized by morphological alterations which also involve the plasma cell organelles. These morphological episodes are epitomized by expression of phosphatidylserine residues on the external surface of the plasma membrane due to flipping of the membrane in early apoptosis followed by chromatin with unusual generalized condensation culminating in nucleus disintegration and fragmented helical strand-breaks [29,30]. When Annexin V/dead cell assay was performed along with the Acridine Orange/Ethidium Bromide assay, LA-39B and LA-55 were shown to induce an apoptotic type of cancer cell death in A549 cells whereas these characteristics were not observed in the non-cancerous normal Raw 264.7 mammalian cells. Since all four of the quinoxaline derivatives studied contain a common benzopyrazine core ring structure, it is most probable that the differential effects observed are a result of the unique side chains attached. These unique side chains could be associated with the cancer cell death-inducing abilities of 3-(quinoxaline-3-yl) prop-2-ynyl methanosulphate (LA-39B) and 3-(quinoxaline-3-yl) prop-2-yn-1-ol (LA-55).

## 4. Methods 

### 4.1. Ferric Reducing Power Assay (FRAP)

The FRAP assay is principled on the ability of antioxidant-containing compound to react with potassium ferricyanide to form ferrocyanide. This will then react with ferric chloride to form ferrous complex that has absorption maximum at 700 nm. Antioxidant activity of quinoxalines was assessed following the ferric reducing power assay protocol as previously described [18]. Briefly, 100 µL of quinoxaline derivatives and ascorbic acid were prepared at various concentrations and 0.2M phosphate buffer (250 µL) and 1% potassium ferricyanide (250 µL) were added to the 100 µL of quinoxaline derivatives, ascorbic acid, or water and the 600 µL mixture was incubated for 20 min at 50 °C. To the mixture, 10% trichloroacetic acid (250 µL) was added and vortexed for 30 s. From the mixture, 250 µL was aliquoted into a new test tube and freshly prepared 0.1% ferric chloride (250 µL) was added. Absorbance was measured at 700 nm using a GloMax-Multi microplate reader (Promega, USA). To calculate the percentage of reducing power the following formula was used: $\%\;increase\;in\;reducing\;power=\frac{A_{test}}{A_{blank}}-1\times 100$.

### 4.2. Quantitative DPPH Assay

The principle of the assay is based on the ability of antioxidant containing compounds to scavenge the DPPH stable free radical by reducing the purple colored DPPH to yellow color. The assay was carried out as described previously [19]. Briefly, 300 µL of various concentrations of quinoxaline derivatives were prepared at various concentrations. Ascorbic acid was used as a standard with correlating concentrations to quinoxalines and water as a blank. For assaying, 100 µL of 0.4 mM DPPH in absolute methanol was then added and incubated for 30 min in the dark. Absorbance was then measured at 517 nm using GloMax-Multi microplate reader (Promega, Madison, WI, USA). To calculate the percentage of inhibition the following formula was used:(1)$$\%\;Inhibition=\frac{A_{blank}-A_{test}}{A_{blank}}\times 100$$

### 4.3. Cell Culture

Raw 264.7 macrophage cells, lung cancer A549 cells, breast cancer cells MCF-7, and cervical cancer HeLa cells were maintained at 37 °C, in a humidified 95% air and 5% CO_2_ environment in cell culture flasks. Raw 264.7 cells and A549 cells were maintained in RPMI-1640 culture medium supplemented with 10% foetal bovine serum (FBS), 2 mM L-glutamine and 1x penicillin, streptomycin, and neomycin mixture (PSN). Cervical cancer HeLa and breast cancer MCF-7 cells were maintained under the same conditions but maintained in Dulbecco’s modification of Eagle medium (DMEM) medium supplemented with 10% FBS, 2 mM L-glutamine, and 1× PSN. Cell density was determined by diluting cells with 10× trypan blue dye and counted using a Life technologies countess II FL automated cell counter (Waltham, MA, USA).

### 4.4. MTT Assay

Cell viability was determined using MTT [3-(4,5-Dimethylthiazol-2-yl)-2,5-diphenyltetrazolium Bromide] assay which is principled on the ability of mitochondrial succinyl dehydrogenase in viable cells to convert soluble yellow tetrazolium salt into an insoluble formazan product. Briefly, cells (Raw 264.7, A549, HeLa, or MCF-7) were seeded at a density of 6 × 10^4^ pre-well in a 96 well-plate and incubated overnight. Cell were then treated with various concentrations of quinoxaline derivatives (25 µM, 50 µM, 100 µM, and 200 µM), 0.5% DMSO in culture medium and 20 μg/mL Actinomycin-D (Sigma Aldrich, Saint Louis, MO, USA) for 24 hrs. Prior to the addition of the MTT reagent, cell imaging was conducted. MTT of 5 mg/mL (Sigma Aldrich, Saint Louis, MO, USA) was added and after 4 h of incubation the aqueous medium was replaced with 100 µL DMSO. The blue formazan crystals were allowed to dissolve in DMSO by incubating in the dark for 30 min. Absorbance was then measured at 570 nm using GloMax-Multi microplate reader (Promega, Madison, WI, USA). 

### 4.5. Analysis of ROS Production

This fluorescence-based assay is principled on the ability of the H_2_DCFDA dye to permeate the plasma membrane and fluoresce to 2’,7’-dichlorofluorescein (DCF) upon encounter with ROS after removal of the diacetate moiety by cellular esterases. Briefly, cells were seeded in 12 well plates at a density of 6 × 10^5^ cells/mL for 24 h. Oxidative stress was stimulated in Raw 264.7 cells using Lipopolysaccharides (10 µg/mL) for 24 h and in A549 cells using hydrogen peroxide (1 mM) for 1 h before staining. Cells were then treated with quinoxaline derivatives at 25 µM and Actinomycin D 20 µg/mL for 24 h. Cells were detached, centrifuged, and re-suspended in culture medium with H_2_DCFDA for 30 min at 37 °C in the dark. Analysis was then carried out using the Muse™ Cell Analyzer.

### 4.6. Apoptosis Analysis using Acridine Orange and Ethidium Bromide

The extent of apoptosis in cells was assayed using the Acridine Orange/Ethidium Bromide assay. This assay is based on staining cells with Acridine Orange (AO) which stains the DNA and RNA of live cells and Ethidium Bromide (EB) which stains exposed DNA and RNA of dead/necrotic cells [28]. The combination of the two dyes represents a measure apoptosis. Briefly, cells were seeded at a density of 6 × 10^5^ cells/mL on coverslips in 6 well-plates overnight. Cells were then treated for 24 h with quinoxaline derivatives at 25 µM and Actinomycin D at 20 µg/mL Plates were centrifuged at 5000 rpm for 5 min then washed with PBS. Cells were then fixed with 2.7% paraformaldehyde for 30 min and centrifuged for 5 min at 5000 rpm. Cells were stained with Acridine Orange 100 µg/mL for 5 min, centrifuged and washed then stained with Ethidium Bromide 100 µg/mL for 5 min and washed again after centrifuging. Coverslips were then mounted onto slides and viewed under the microscope using the Nikon Ti-E inverted microscope.

### 4.7. Annexin V and Dead Cell Apoptosis Analysis

This assay is principled on Annexin V and PI stain which is based on the staining of cells undergoing early apoptosis with Annexin V which binds to phosphatidylserine which is being expressed on an impermeable membrane, while with late apoptosis, PI will be able to bind to the DNA due to permeability of the membrane. Muse^®^ Annexin V and Dead Cell Assay kit was used according to manufacturer’s instruction. Briefly, cells were seeded in 12 well-plates at a density of 6 × 10^5^ cells/mL for 24 h. They were then treated with quinoxaline derivatives at 25 µM and Actinomycin D 20 µg/mL for 24 h. Cell were detached, centrifuged, and resuspended in 1% FBS. Muse™ Annexin V and Dead Cell Reagent was then added to the suspension and incubated for 20 min in the dark. Analysis was then carried out using the Muse™ Cell Analyzer.

### 4.8. Designing the Quinoxaline Derivatives

In this study, the four quinoxaline derivatives were designed to possess the signature benzopyrazine core molecule made of a benzene ring and a pyrazine ring capable of triggering antimicrobial activities and induce anticancer properties due to their ability to bind DNA [31,32]. Four slight variations were made on the side chains attached to the benzopyrazine core group. In the first quinoxaline derivative designed (Figure 1), 3-(quinoxalin-3-yl)prop-2-yn-1-ol (LA-55), the prop-2-nl-1-ol side chain was incorporated as in several anticancer agents wherein anti-proliferative effects have been observed [33] by targeting microRNA-21 and reversing phenotypes in cancer cells [34]. In the second quinoxaline derivative (Figure 2), 2-methyl-4-(quinoxalin-3-yl)but-3-yn-2-ol (LA-65C3), the 2-methyl-4-but-3-yn-2-ol group is attached to the benzopyrazine core. The 2-methyl-4-but-3-yn-2-ol group has been shown to inhibit Akt kinase and exhibit antitumor activities [35]. In the third quinoxaline derivative (Figure 3), 3-(quinoxalin-3-yl)propionaldehyde (LA-16A), a propanal or propionaldehyde group is attached to the benzopyrazine core group. Propanal groups are known to carry potent anti-inflammatory activities [36]. In fourth quinoxaline derivative (Figure 4), 3-(quinoxalin-3-yl)prop-2-ynyl methanesulfonate (LA-39B), a methanesulfonate group is attached. Methanesulfonates are known to cause hyperacetylation of histone deacetylase inhibitors thereby increasing susceptibility to cancer cell death [37]. The four variations made on the side chains attached to the quinoxaline core structure are aimed at searching for more potent forms of the envisaged candidate molecules. The mode of action of each of the quinoxaline derivatives is yet to be determined as these are newly synthesized novel molecules.

### 4.9. Statistical Analysis

All experiments were carried out in duplicate and repeated three times to confirm consistency and determine level of variation. Data was reported as mean ± SD and statistical analysis was performed using Graph Pad Instat^TM^ 3 software with variants test (ANOVA), Tukey–Kramer to compare samples. Each value represents the mean ± SD of three experiments performed in triplicate wherein p-value represents the level of significant changes and *p* < 0.05 is considered statistically significant. Levels of significance are denoted using an associated asterix as follows: * *p* < 0.05, ** *p* < 0.01, and *** *p* < 0.001.

## 5. Conclusions

The results obtained demonstrate that quinoxaline derivatives, 3-(quinoxaline-3-yl) prop-2-ynyl methaonosulphate (LA-39B) and 3-(quinoxaline-3-yl) prop-2-yn-1-ol (LA-55), can potently induce significant anticancer properties against A549 lung cancer cells while displaying no cytotoxicity against the non-cancerous Raw 264.7 macrophage cells. These newly developed series of quinoxaline derivatives may present new alternatives with potential abilities for use in lung cancer treatment.

## Figures and Tables

**Figure 1 molecules-24-00407-f001:**
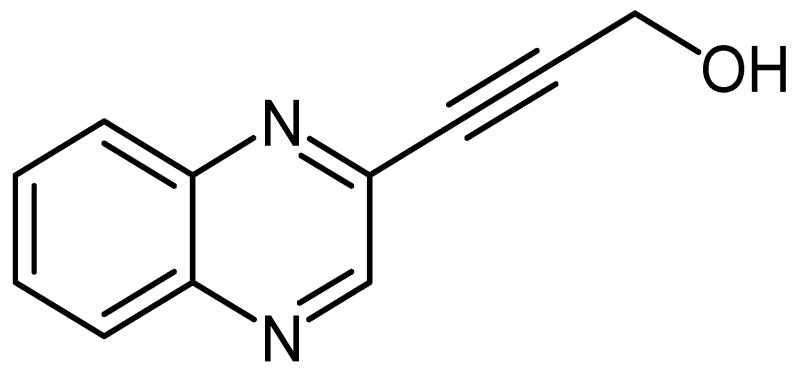
The chemical structure of 3-(quinoxalin-3-yl)prop-2-yn-1-ol (LA-55).

**Figure 2 molecules-24-00407-f002:**
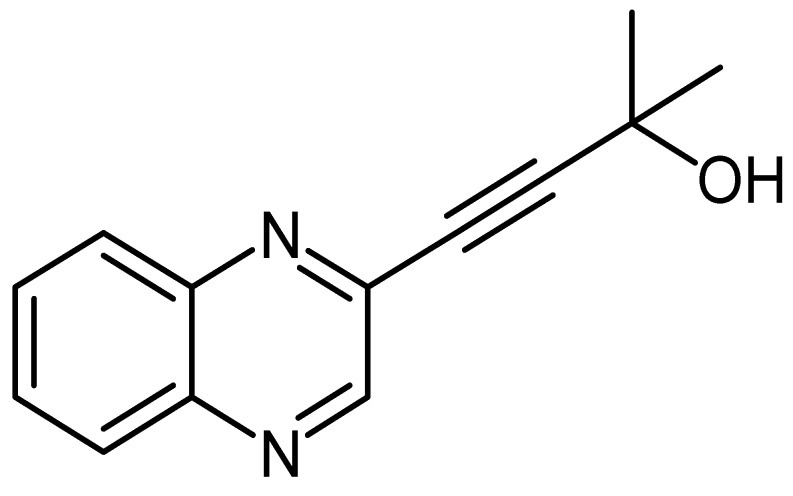
The chemical structure of 2-methyl-4-(quinoxalin-3-yl)but-3-yn-2-ol (LA-65C3).

**Figure 3 molecules-24-00407-f003:**
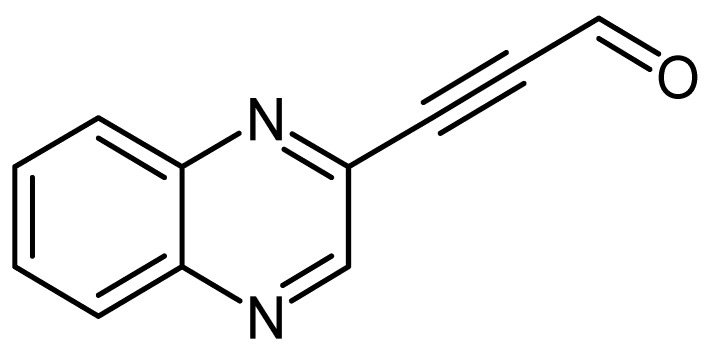
The chemical structure of 3-(quinoxalin-3-yl)propionaldehyde (LA-16A).

**Figure 4 molecules-24-00407-f004:**
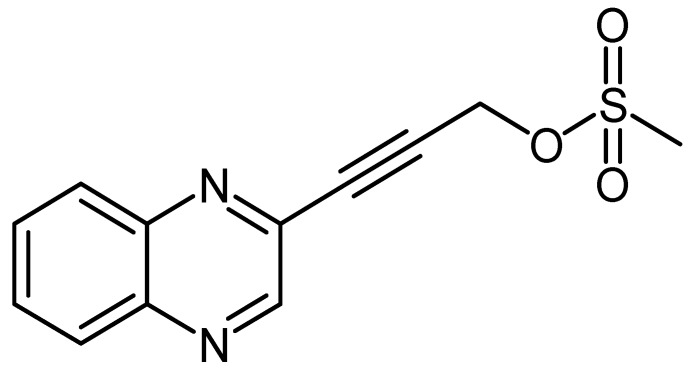
The chemical structure of 3-(quinoxalin-3-yl)prop-2-ynyl methanesulfonate (LA-39B).

**Figure 5 molecules-24-00407-f005:**
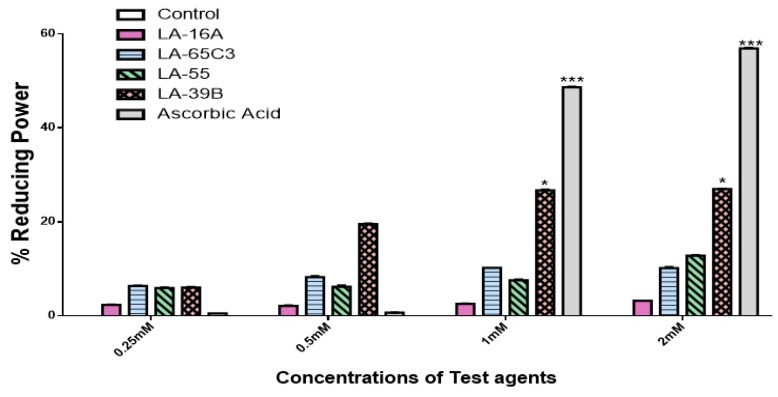
Reducing power potential of quinoxaline derivatives. The reducing power potential of quinoxaline derivatives LA-39B, LA55, LA-65C3, and LA-16A was assayed at various concentrations (0.25 to 2mM), using the FRAP assay with ascorbic acid as a standard and water as control. Each value represents the mean ± SD of three experiments performed in triplicates independently. (* *p* < 0.05 and *** *p* < 0.001).

**Figure 6 molecules-24-00407-f006:**
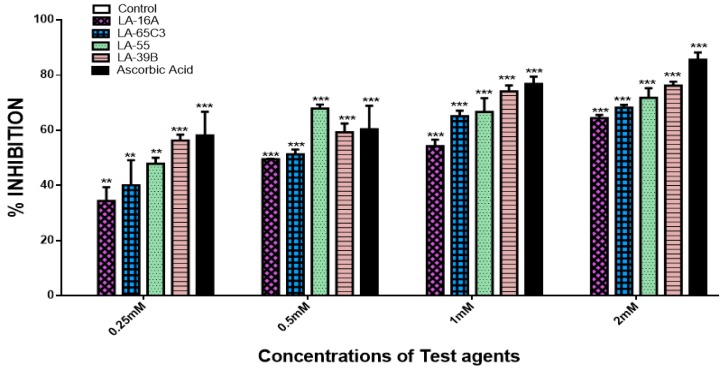
Free radical scavenging properties of quinoxaline derivatives. The free radical scavenging activities of quinoxaline derivatives LA-39B, LA55, LA-65C3, and LA-16A were assayed at various concentrations (0.25–2 mM) using the DPPH assay with ascorbic acid as a standard and water as control. Each value represents the mean ± SD of three experiments performed in triplicates independently. (** *p* < 0.01 and *** *p* < 0.001).

**Figure 7 molecules-24-00407-f007:**
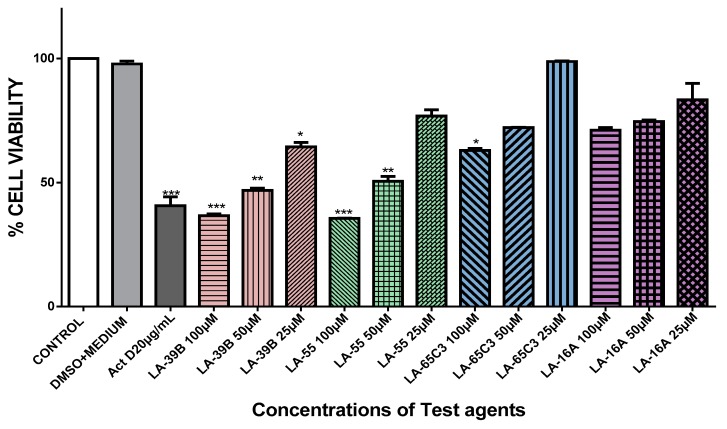
The effect of quinoxaline derivatives on cell viability of HeLa cervical cancer cells. Cell viability of HeLa cells when treated with quinoxaline derivatives LA-39B, LA55, LA-65C3, and LA-16A at various concentrations (25 to 100 µM) was assayed using the MTT assay. Actinomycin D (20 µg/mL) was used as a positive control and DMSO-treated cells as controls. Each value represents the mean ± SD of three experiments performed in triplicates independently. (* *p* < 0.05, ** *p* < 0.01, and *** *p* < 0.001).

**Figure 8 molecules-24-00407-f008:**
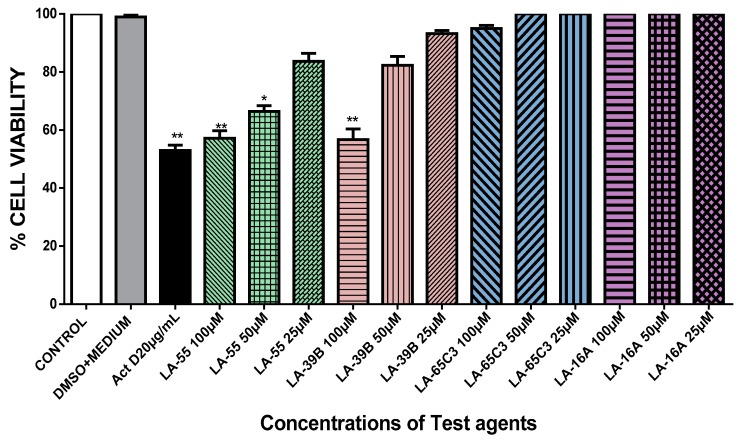
The effect of quinoxaline derivatives on cell viability of MCF-7 breast cancer cells. MCF-7 cells were treated with quinoxaline derivatives LA-39B, LA55, LA-65C3, and LA-16A at various concentrations (25 to 100 µM) for 24 h and cell viability determined via the MTT assay. Actinomycin D (20 µg/mL) was used as a positive control and DMSO-treated cells as negative controls. Each value represents the mean ± SD of three experiments performed in triplicates independently. (* *p* < 0.05 and ** *p* < 0.01).

**Figure 9 molecules-24-00407-f009:**
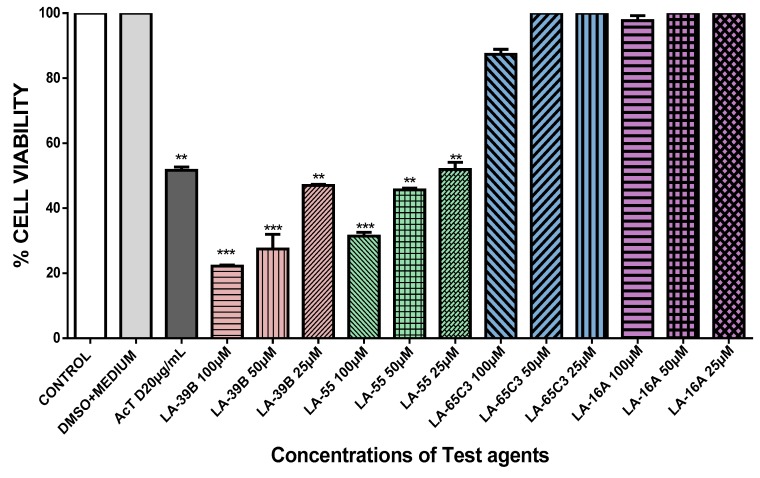
The effect of quinoxaline derivatives on cell viability of A549 lung cancer cells. A549 cells were treated with quinoxaline derivatives LA-39B, LA55, LA-65C3, and LA-16A at various concentrations (25 to 100 µM) for 24 h and cell viability determined via the MTT assay. Actinomycin D (20 µg/mL) was used as a positive control and DMSO-treated cells as negative controls. Each value represents the mean ± SD of three experiments performed in triplicates independently. (** *p* < 0.01 and *** *p* < 0.001).

**Figure 10 molecules-24-00407-f010:**
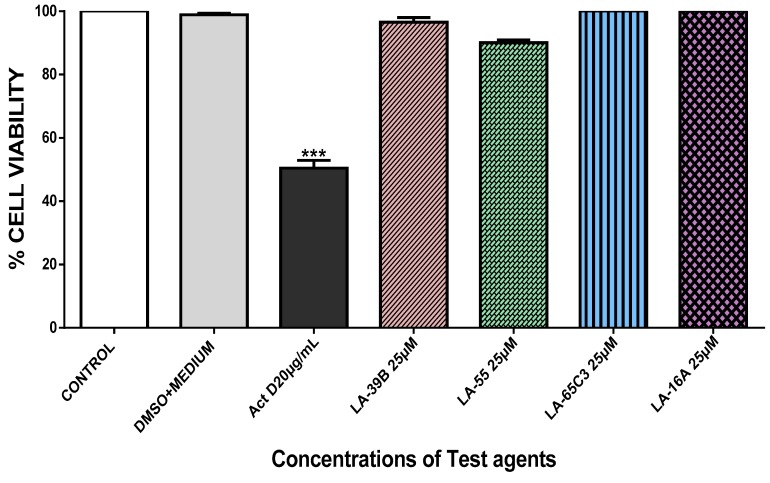
The effect of quinoxaline derivative on cell viability of Raw 264.7 cells. Cell viability of Raw 264.7 cells after treatment with quinoxaline derivatives LA-39B, LA55, LA-65C3, and LA-16A at 25 µM was assayed via the MTT assay using Actinomycin D (20 µg/mL) as a positive control and DMSO-treated cell as a negative control. Each value represents the mean ± SD of three experiments performed in triplicates independently. (*** *p* < 0.001).

**Figure 11 molecules-24-00407-f011:**
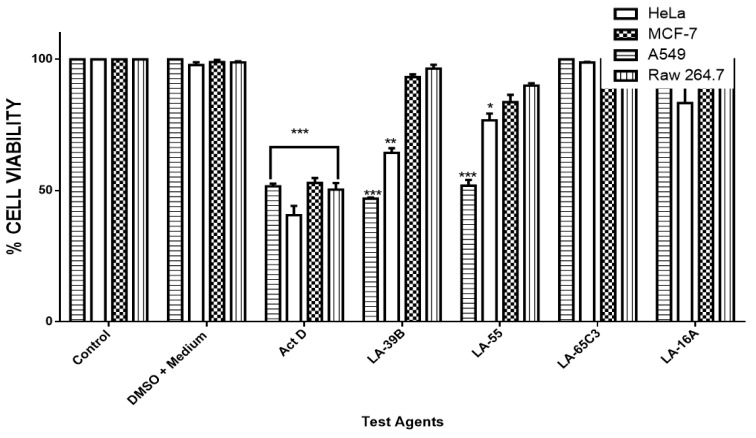
The comparative effect of quinoxaline derivatives on cell viability of HeLa, MCF-7, A549, and Raw 264.7 cells. Cell viability of HeLa, MCF-7, A549, and Raw 264.7 cells when treated with quinoxaline derivatives LA-39B, LA55, LA-65C3, and LA-16A at 25µM was assayed via the MTT assay using Actinomycin D(20 µg/mL) as a positive control and DMSO-treated cell as negative controls. Each value represents the mean ± SD of three experiments performed in triplicates independently. (* *p* < 0.05, ** *p* < 0.01, and *** *p* < 0.001).

**Figure 12 molecules-24-00407-f012:**
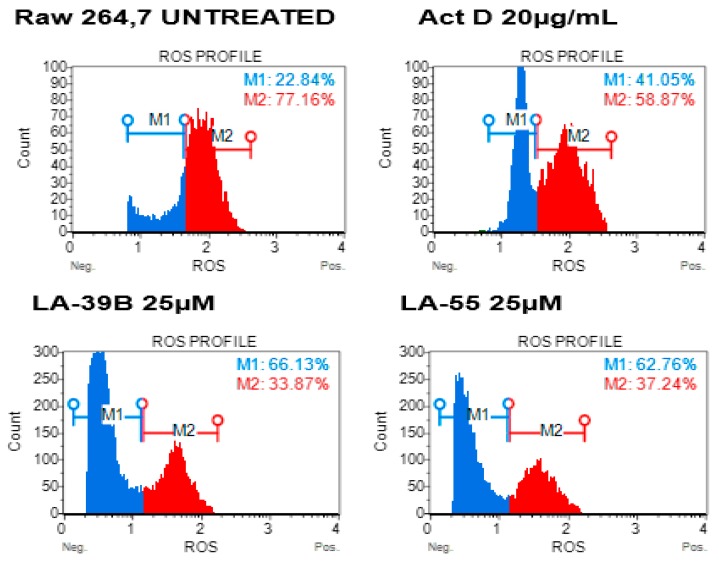
The effect of quinoxaline derivatives LA-39 and LA-55 on reactive oxygen species (ROS) production in Raw 264.7 cells. Raw 264.7cells were stimulated with LPS and treated with quinoxaline derivatives LA-39B and LA55 at 25µM. Flow-cytometry analysis was carried out using the Muse^TM^ Cell Analyzer.

**Figure 13 molecules-24-00407-f013:**
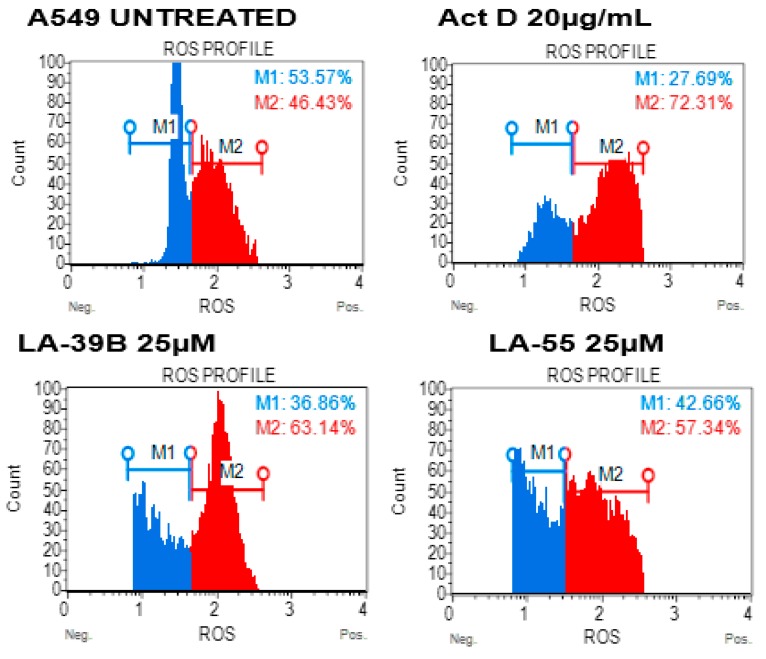
The effect of quinoxaline derivatives LA-39 and LA-55 on ROS production in A549 cells. Raw 264.7cells were stimulated 1mM H_2_O_2_ and treated with 25µM of quinoxaline derivatives LA-39B and LA55. Actinomycin D (20 µg/mL) was used as a positive control. Flow-cytometry analysis was carried out using the Muse^TM^Cell Analyzer.

**Figure 14 molecules-24-00407-f014:**
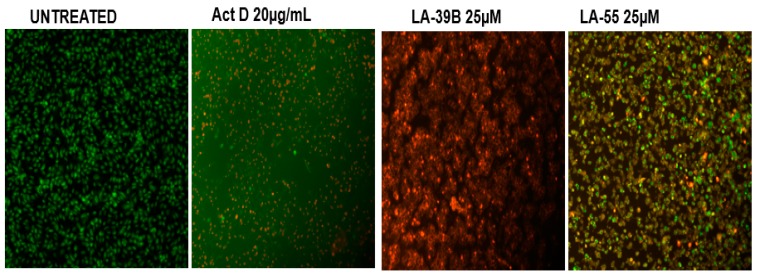
Apoptosis evaluation in A549 cells using AO/EB fluorescence microscopy. Fluorescence of A549 cells when treated with 25 µM quinoxaline derivatives LA-39B and LA55 was assayed using Actinomycin D (20 µg/mL) as a standard and untreated cell as control. Images were captured using the Nikon Ti-E inverted microscope.

**Figure 15 molecules-24-00407-f015:**
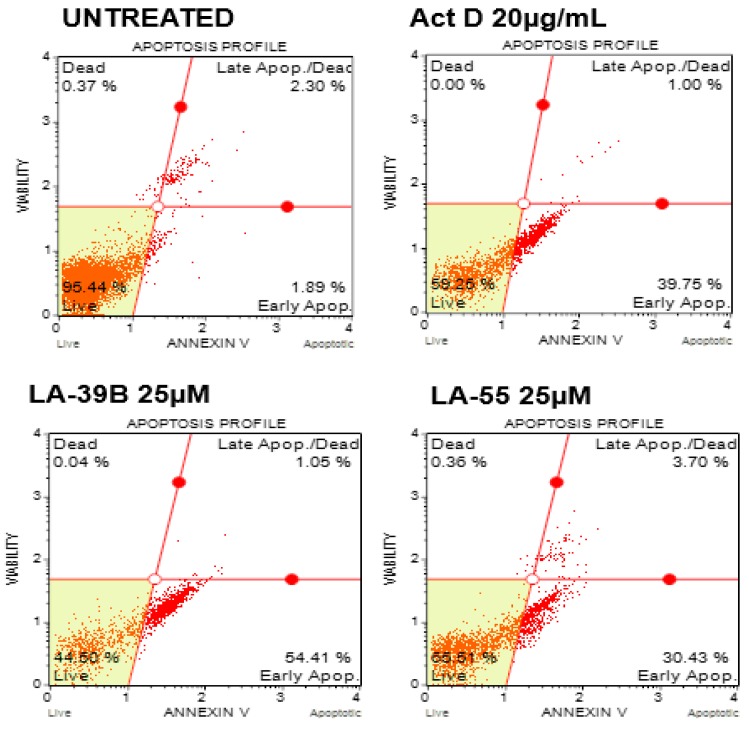
Evaluation of apoptosis induction in A549 cells using the Annexin V/dead cell assay. The A549 cells were treated with 25 µM of quinoxaline derivatives (LA-39B and LA55) for 24 h and apoptosis was assayed via flow cytometry using the Annexin V/dead cell assay. Actinomycin D was used a positive control with untreated cells used as negative controls. Flow cytometry was carried out using the Muse^TM^Cell Analyzer.

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
