# Peer review of "Induction of Cell Death in Human A549 Cells Using 3-(Quinoxaline-3-yl) Prop-2-ynyl Methanosulphonate and 3-(Quinoxaline-3-yl) Prop-2-yn-1-ol"

_molecules, 2019, doi:10.3390/molecules24030407_

Round 1
Reviewer 1 Report
This manuscript conducted studies on the synthesis of 4 compounds and their anticancer activity. From the chemistry of view, there is no novelty, most importantly, there are no basis for molecular designation description, target based novel drug development is of importance. From the chemistry of view, the authors conducted a relatively complete studies, however, the authors chose actinomycin D as a positve control without discuss the reasons, there are lots of anticancer drugs with very good activity, I suggest use more positive controls with various mode of action when the compounds exibit good activity if the authors did not provide studies on the mode of action at target probing.
English need improve, for example the last sentence used drug, it is not proper becasue drug is the compound registered.
From the above opinion, I suggest reject this manuscript for publication.
Author Response
RE: RESPONSE TO COMMENTS OF REVIEWER NO. 1
Dear Editor
Reviewer No. 1 made the following points and responses follow each comment made:
Reviewer 1
This manuscript conducted studies on the synthesis of 4 compounds and their anticancer activity. From the chemistry of view, there is no novelty, most importantly, there are no basis for molecular designation description, target based novel drug development is of importance. From the chemistry of view, the authors conducted a relatively complete studies, however, the authors chose actinomycin D as a positve control without discuss the reasons, there are lots of anticancer drugs with very good activity, I suggest use more positive controls with various mode of action when the compounds exibit good activity if the authors did not provide studies on the mode of action at target probing.
I think most of these sentiments were echoed by reviewer 2 and we have responded accordingly to them. Novelty of the drugs: The review comments there is no novelty; to our knowledge there is no published data on these drugs or their synthesis. They are novel drugs. We have also added more information on what guided the designs. We chose actinomycin because it served the purpose of being the positive control. Yes, we could have used other alternatives but Act-D served the purpose. We cannot provide the mode of action of these drug derivatives at this point, it is our next step.
English need improve, for example the last sentence used drug, it is not proper becasue drug is the compound registered.
We have replaced the word “drugs” with more specific “quinoxaline derivatives” as adviced in line 393-394. We would have wished the generalised comment on improving the writing could have been more guiding but we have duly taken the responsibility of making further reviews.
We added the word “potential”…line 55.
Extended revisions were made as indicated on the “Track Changes” word form.
We hope that the responses are satisfactory.
Kindly
Prof TM Matsebatlela
BMBT

Reviewer 2 Report
Molecules-414777
The authors synthesized several compounds as a cohort of quinoxaline derivatives and suggested that they are potential candidate drugs through several bioactivities assay.
I have some comments to this manuscript.
Major points:
This manuscript is lack of focus to the readers.
1. Synthesis part:
The authors synthesized a series of compounds. How can they design to synthesize it? What is their molecular target? Where is docking data? What is really target?
2. Biology part:
The authors evaluated with various different bioassays and suggested some conclusions.
2.1 The authors should explain a bit about their each bioassays and make logical explanation between each bioassay part.
2.2 As shown in the manuscript, there are several potential activities such as cell viability, free radical scavenging, oxidative stress… If that, how about the selective effects of their compounds?
3. Line 362 page 14. “can potently induce significant anticancer properties against A549 lung cancer cells while displaying no cytotoxicity against the non-cancerous RAW267.7 macrophages cells”
In fact, RAW264.7 cells are macrophages cells (no non-cancerous lung cells). It will be more valuable if the authors will also check with non-cancerous lung cells to make this conclusion.
Minor points:
1. The authors should edit some minor mistakes in their manuscript.
For example:
Line 125, page 4: “figure 3.1”.
Line 233, page 10: mistake fonts
Line 322, page 13: “A.nalysis”
2. The authors also used p-value with (*) and (**), please indicate the value in the manuscript also, especially in legends of figures as it appears.
3. Fonts are not consistent, for example: in Figure 7 vs Figure 8.
4. Statistical analysis part: the authors should indicate p-value here also.
Author Response
RE: RESPONSE TO COMMENTS OF REVIEWER NO. 2
Dear Editor
Reviewer No. 2 made the following Major points and responses follow each comment made:
This manuscript is lack of focus to the readers.
1. Synthesis part:
The authors synthesized a series of compounds. How can they design to synthesize it? What is their molecular target? Where is docking data? What is really target?
To respond to this comment we have attached section 4.8(lines 362-381) to explain the idea behind the drug design. We hope this offers a more elaborate explanation as to how the design was strategized. These are novel synthetics; hence, the mode of action is yet to be determined. The idea here was to show that by slightly varying the composition of the side chains we were able to obtain four different outcomes and we continue now to determine their mode of action as envisaged from other studies that used such side chains. The novel nature of the quinoxaline derivatives compels us to delineate their mode of action to accompany the observed results. New references added on lines 462-485 to accompany the newly inserted information.
2. Biology part:
The authors evaluated with various different bioassays and suggested some conclusions.
2.1 The authors should explain a bit about their each bioassays and make logical explanation between each bioassay part.
I am not sure I understand clearly what is required from this review. We have provided the principles of all assays in Section 4 (4. Methods) and continued to link the principles with the outcomes in the discussion. I am not sure on how to provide “logical explanation between each bioassay part” and where this will be inserted in the manuscript. Unless if the view is that the principles are not sufficiently clear.
2.2 As shown in the manuscript, there are several potential activities such as cell viability, free radical scavenging, oxidative stress… If that, how about the selective effects of their compounds?
To determine the selective nature of the derivatives, we have provided a non-cancerous cell line, a macrophage cell line, as a control cell line to show that the quinoxaline derivatives under study are not toxic to cells of the immune system that are mostly active during the cancerous activities. The activity status of the macrophages can either inhibit or promote growth of the cancer cells, hence, our choice to use macrophages as control cells. However, we agree that using non-cancerous lung cancer cells would also provide a side-by-side comparison to lung cancer cells. As we continue to investigate the effect of these derivatives on various other cancer cell lines, we would have to get a more specific non-cancer cell line for each cancer of the cancer cell types.
3. Line 362 page 14. “can potently induce significant anticancer properties against A549 lung cancer cells while displaying no cytotoxicity against the non-cancerous RAW267.7 macrophages cells”
In fact, RAW264.7 cells are macrophages cells (no non-cancerous lung cells). It will be more valuable if the authors will also check with non-cancerous lung cells to make this conclusion.
We agree with the reviewer, hence, we don’t make any claim that the drug is selective against normal LUNG cells but non-cancerous macrophages. For us to make a claim of selectivity between cancerous and non-cancerous lung cells we would have to acquire a more suitable non-cancerous subtype of A549 cells and design parallel toxicity assays. However, since our focus pointed us towards inflammation-based phenomenon observed through preliminary assays, we resorted to using macrophages as the central role player in inflammation and most active anti-tumor cells of the immune system. We therefore confine our claim to non-cancerous cells and not non-cancerous lung cancer cells.
Minor points:
1. The authors should edit some minor mistakes in their manuscript.
For example:
Line 125, page 4: “figure 3.1”. Corrected to “figure 5” as per advice.
Line 233, page 10: mistake fonts. The font mistake has been correct.
Line 322, page 13: “A.nalysis”. Changed to “Analysis” by removing the dot.
2. The authors also used p-value with (*) and (**), please indicate the value in the manuscript also, especially in legends of figures as it appears.
The description of the p-values is added on lines 386-389. Each of the figure legends p-values have been differentially denoted as per advice.
3. Fonts are not consistent, for example: in Figure 7 vs Figure 8.
I am failing to determine if the comment is on font type or size. The only difference I can spot between figure 7 and 8 is that figure 8 is bigger than figure 7 but the scale should be able to ease out the difference. But I am not sure this is the difference referred herein.
4. Statistical analysis part: the authors should indicate p-value here also.
The description of the p-values is duly added on lines 386-389.
Kindly
Prof TM Matsebatlela
BMBT

Reviewer 3 Report
The manuscript describes the synthesis and biological evaluation of a few number of quinoxaline derivatives with potential anti-proliferative activity.
There are some points thau must be reviewed:
Please control the 1H NMR spectrum of compound LA55
Why did you choose these three concentrations? It Will be useful to create a dose response curve
Why Act-D was used at this dose?
Please add in the introduction these references to improve the quality of your design:
Wei Lv et al., 2016 J Med Chem 59,10, 4511-4525
F. Aiello et al., 2017, ChemMedChem 12,16, 1279-1285.
Author Response
RE: RESPONSE TO COMMENTS OF REVIEWER NO. 3
Dear Editor
Reviewer No. 3 made the following points and responses follow each comment made:
Reviewer 3
The manuscript describes the synthesis and biological evaluation of a few number of quinoxaline derivatives with potential anti-proliferative activity.
There are some points thau must be reviewed:
Please control the 1H NMR spectrum of compound LA55
I am not I understand what type of response this statement required.
Why did you choose these three concentrations? It Will be useful to create a dose response curve
I am of the view that it is clearly indicated that the cell viability studies guided the choices of the dosages as the study proceeds. Unless I get a more clearer review.
Why Act-D was used at this dose?
Act-D inhibits cell growth at this dosage as determined in previous studies. This therefore makes it a good positive control at this dosage to show that cells are not resistant to drug-induced cell death. We are not comparing Act-D to our drug derivatives but simply showing that our chosen cells lines are not engineered in any way to be resistant to cell death, especially against the non-cancerous macrophages.
Please add in the introduction these references to improve the quality of your design:
Wei Lv et al., 2016 J Med Chem 59,10, 4511-4525
F. Aiello et al., 2017, ChemMedChem 12,16, 1279-1285.
We have added information from the two references as per advice (lines 45-50) and included the reference in the reference list (lines 410-414).
Kindly
Prof TM Matsebatlela
BMBT

Round 2
Reviewer 1 Report
most of the point are revised, suggest accept this manuscript for publish before minor revision, but English need revise again, for example:.
1.LRMS suggest changed into MS.
2."µg\mL" changed into "µg/mL"
Author Response
We have effected both suggested changes and continued to revise the entire manuscript in accordance with the comment of Reviewer 1.
1.LRMS suggest changed into MS. (line 88)
2."µg\mL" changed into "µg/mL". (lines 354, 356, 365, 378).
Additional revisions were made as indicated on the “Track Changes” word form.
We hope that the responses are satisfactory.
Kindly
Prof TM Matsebatlela
BMBT

Reviewer 2 Report
molecules-414777
It will be better if the authors can shortly introduce function of macrophages cells in their Introduction part since evaluating on RAW264.7 cells is also important point in this manuscript.
Author Response
We have duly added this part as per advice. (Line 61-69)
Since the reviewer mad a comment of English minor revisions and research design descriptions, we have revised the manuscript taking these comments into consideration. The minor revisions can be followed on the “track changes” word document.
Kindly
Prof TM Matsebatlela
BMBT
